# Dual-head Genre-instance Transformer Network for Arbitrary Style Transfer

## ABSTRACT

Arbitrary style transfer aims to render artistic features from a style reference onto an image while retaining its original content. Previous methods either focus on learning the holistic style from a specific artist or extracting instance features from a single artwork. However, they often fail to apply style elements uniformly across the entire image and lack adaptation to the style of different artworks. To solve these issues, our key insight is that the art genre has better generality and adaptability than the overall features of the artist. To this end, we propose a Dual-head Genre-instance Transformer (DGiT) framework to simultaneously capture the genre and instance features for arbitrary style transfer. To the best of our knowledge, this is the first work to integrate the genre features and instance features to generate a high-quality stylized image. Moreover, we design two contrastive losses to enhance the capability of the network to capture two style features. Our approach ensures the uniform distribution of the overall style across the stylized image while enhancing the details of textures and strokes in local regions. Qualitative and quantitative evaluations demonstrate that our approach exhibits its superior performance in terms of visual qualitative and efficiency.

## CCS CONCEPTS

• **Applied computing** → **Fine arts**; • **Computing methodologies** → **Appearance and texture representations**.

## KEYWORDS

Arbitrary style transfer, dual-head style learning, contrastive learning

## 1 INTRODUCTION

If great artworks tell a complete story, the artistic style acts as its essence, unveiling the thematic core and unique creative perspective of the composition. The objective of Arbitrary Style Transfer (AST) is to transfer the style of a reference image to any given image while preserving its content [9]. Given its great potential in various practical applications, AST has emerged as a prominent research area in computer vision, garnering considerable attention to driving continuous endeavors in both quality and efficiency.

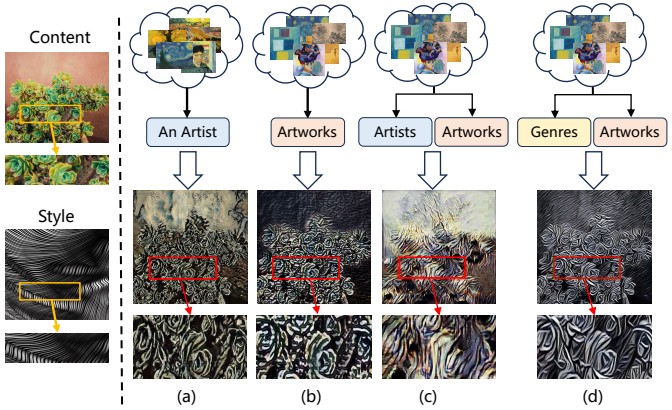

**Figure 1: Learn style representation from different perspectives. (a) Style learning from a specific artist. (b) Style learning from a single artwork. (c) Style learning from certain artists and artworks. (d) Our style learning from genres and artworks.**

The primary challenge of AST lies in effectively representing style and then realistically mapping an image into an artistic rendition. Most existing methods learn style representation using the following two approaches. The first one is to learn the holistic style from a specific artist such as Van Gogh. Prior works [14, 18, 21, 41] have successfully produced high-quality stylized images by treating each artist's style as a domain. However, a common limitation exists: these methods tend to generate only a single type of stylization, making them less adaptable to different artwork styles, as illustrated in Fig. 1 (a). The alternative approach is to abstract the style representation from a single artwork [3, 6, 25, 26, 37, 38]. Such methods focus on extracting instance features such as specific brushstrokes and textures, enabling controllable generation of images. However, they often struggle to maintain uniform style elements across the entire image due to limited utilization of comprehensive style information present in art collections. As mentioned in [18], relying on a single artwork may not adequately represent the entirety of an artistic style. This can lead to issues such as local content leaks and inconsistencies in holistic style, as illustrated in Fig. 1 (b).

Recently, some studies [4, 28, 32] attempt to combine the above two approaches to achieve a more comprehensive style representation. Chen *et al.* propose DualAST [4] to learn the instance features using a pre-trained VGG-19 [19] and artist's style features through GAN-based constraints. Xu *et al.* introduce DRB-GAN [32], leveraging Dynamic ResBlocks to integrate both artist's style and instance features. Wu *et al.* utilize a DSTM [28] to decouple the artist's style and instance style from a single style image and swap them with those from other images to achieve AST. However, these methods are constrained by the scope of their training data, with DualAST

[4] trained on just six artists and DRB-GAN [32] on eleven collected from WikiArt, limiting their capability for arbitrary style transfer. As illustrated in Fig. 1 (c), when presented with an unseen artistic style, the resulting stylized image may exhibit local distortions, compromising the content structure.

In this paper, we propose a novel insight that the ***art genre*** offers greater generality and adaptability compared to the overall features of individual artists. The genre encompasses distinctive visual elements, techniques, and methods, often corresponding to an art movement or school, such as Baroque, Expressionism, or Impressionism. We recognize that an artist's style in the genre evolves significantly over time. For example, Van Gogh's early artwork "Papa Tanguy" shows Ukiyo-e influences, whereas "Starry Night" embodies Post-Impressionism. These genre differences highlight that even the artworks by the same artist can vary significantly in painting technique and style. To this end, we propose the dual-head genre-instance transformer (**DGiT**) framework to enable highly effective AST. Specifically, DGiT consists of a content encoder, a dual-head style encoder, and a style transformer decoder. The dual-head style encoder has two heads, a genre-wise head and an instance-wise head, to simultaneously capture the genre features and instance features. The genre-wise head is designed to capture the common features of the art genre such as the overall feeling, while the instance-wise head is utilized to capture the unique features like colors, texture, and brushstrokes of the artwork. We design an effective style transformer decoder to progressively migrate them to obtain the final style representation. Different from [6, 35] based on ViT, our method not only learns two style representations but also accelerates convergence by the designed style transformer decoder. As shown in Fig. 1 (d), our approach can ensure the style elements are uniformly and coherently applied across the entire image while successfully preserving the overall outline and texture details of the content image.

Furthermore, we introduced two special contrastive losses, namely genre contrastive loss and instance contrastive loss, to enhance the effectiveness of our model. Specifically, the genre contrastive loss treats artworks from the same genre as positive examples and those from different genres as negative, thereby facilitating the learning of holistic genre styles within the same genre. On the other hand, the instance contrastive loss randomly selects positive-negative patch pairs within a single style image and other distinct style images. Unlike existing methods [17, 26, 38, 39] that process the entire image as an anchor to identify overall differences between various artworks, our instance contrastive loss focuses on the texture details and brushstrokes at any spatial location within the style image. Unlike the patch-methods [17, 23, 33] which considers the relationship between the stylized image and the source image, we consider the relationship between a single artwork and other artworks. This contrastive strategy enables the instance-wise head to identify the unique style variations in each image. Our contributions are summarized as follows:

- We propose a dual-head genre-instance transformer (DGiT) framework to simultaneously capture the genre and instance features for achieving arbitrary style transfer. To the best of our knowledge, this is the first work in AST to propose the

combination of the genre-wise and instance-wise features to generate high-quality stylized images.
- Two contrastive losses are introduced to encourage DGiT to capture two style representations, which can strengthen the model's ability of preserving more texture details and the holistic style of the artwork.
- Experimental results show that our approach can achieve the best style performance regarding visual quality. The model allows the style elements to be uniformly and coherently applied across the stylized image, improving the robustness of content-style inputs.

## 2 RELATED WORK

### 2.1 Arbitrary Style Transfer

Since the introduction of adaptive instance normalization for achieving AST in [9], there have been numerous advancements in this area. Some methods [14, 18, 21, 41] focus on learning the holistic style from a specific artist such as Van Gogh. For instance, Svoboda *et al.* proposed a two-stage model [21] for stylized imaging with enhanced content geometry flexibility, while Zhang *et al.* created ArtBank [41] to learn artists' style type by guiding pre-trained large-scale models. Nevertheless, these methods only learn the style type of a limited number of artists and are unable to achieve AST. To address this issue, most existing methods [1, 3, 6, 13, 26, 31, 34, 37, 38] adopt the other perspective to learn the style representation from a single artwork. For example, StyTr2 [6], as a transformer-based approach, mitigates biased content representation in style transfer by accounting for the long-range dependencies in input images. A novel dynamic style kernel [31] performs learning spatial adaptation capabilities to achieve per-pixel stylization. Recently, some advanced methods [15, 37] studies on diffusion models to improve the quality of the stylized image. InST [37] learns the artistic style directly based on the inversion method and DiffuseIT [15] employs a pre-trained ViT model to guide the generation process of DDPM models [8]. Despite these progresses, they fail to apply style elements uniformly due to the limited use of comprehensive style information from art collections. Several studies [4, 28, 32] attempt to merge the two perspectives to achieve a more effective style representation. DualAST [4] utilizes a fixed pre-trained VGG-19 [19] to learn instance features and incorporates multiple artist's style features via GAN constraints. DRB-GAN [32] integrates both artist's style and instance features using Dynamic ResBlocks. However, the limited artist style type constrains the flexibility and generalization of arbitrary style transfer.

Unlike the above methods, we focus on the art genre features which have better generality and adaptability than the holistic features of the artist. Our method can learn simultaneously the holistic genre features and specific artworks' instance features, to achieve high quality and flexible stylized results in AST.

### 2.2 Contrastive Learning for Style Transfer

With the rise of contrastive learning, many studies [2, 23, 26, 29, 33, 38, 40] have investigated contrastive learning in style transfer tasks. Inspired by CUT [17], IEST [17] calculates the feature statistics (mean and standard deviation) as style priors, and confines the calculation of contrastive loss to the generated outputs.

Dual-head style encoder

Content encoder

Style transformer decoder

**Figure 2: Overview pipeline of our DGiT framework. We split the content image $I_c$ and style image $I_s$ into patches, and obtained the responding features $F_c$ and $F_s$ by transformer encoders. A dual-head style encoder simultaneously extract the genre feature $F_{gen}$ and instance feature $F_{ins}$ from the style features $F_s$. A style transformer decoder then progressively fuse the content sequence according to two style representations $F_{gen}$ and $F_{ins}$. The genre contrastive loss $\mathcal{L}_{gen}$ and instance contrastive loss $\mathcal{L}_{ins}$ are utilized to guide the dual-head style encoder to learn the related style features.**

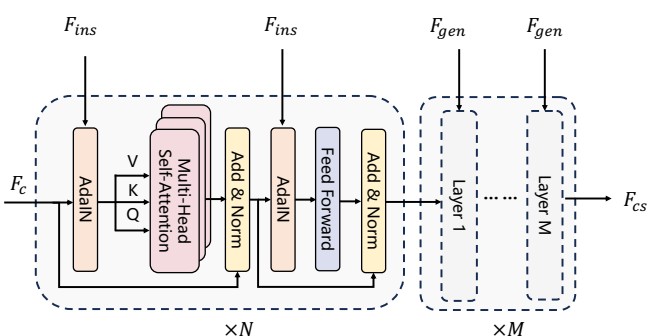

**Figure 3: Style transformer decoder includes AdaIN, multi-head attention, feed-forward network, and layer normalization. We first fuse the content feature $F_c$ with the instance feature $F_{ins}$ and then fuse it with the genre features $F_{gen}$.**

CLAST [23] employs a similar form to quantify the color and style differences between the generated results and the style reference image. ZeCon [33] effectively preserves the content information in a zero-shot manner by leveraging the patch-wise contrastive loss [17]. Unlike the above patch-based methods, CAST [38] uses extracted holistic style features by the pre-trained VGG-19 layers to compute contrastive loss between images. UCAST [39] further uses an adaptive temperature mechanism to control the proportion of penalties between the positive and negative samples. CCPL [29]

introduces contrastive learning for video style transfer by considering the frame-wise patch differences. MicroAST [26] constructs the stylized image with other style images as positive/negative pairs to enhance the capability of the style encoder. CSACT [40] adopts the Gram matrix as a style representation in contrastive learning to capture more style information.

In this paper, we adopt supervised contrastive learning [11] to learn the common features of the art genre. An instance contrastive loss is designed to capture the instance features of the style image. Compared with [26, 27, 38] which uses the full image as the sample, our instance contrastive loss focuses on the texture details and brushstrokes at any spatial location within the same artwork.

## 3 METHOD

In this section, we will introduce our Dual-head Genre-instance Transformer (DGiT) framework for arbitrary style transfer. We first present the overall pipeline of our DGiT framework, followed by the dual-head style encoder, style transformer decoder, and contrastive learning strategy in detail.

### 3.1 Overall Framework

Given a content image $I_c$ and a style image $I_s$, we aim to synthesize an image $I_g$ that has the style pattern of $I_s$ while maintaining the content structure of $I_c$. We propose a dual-head genre-instance transformer (DGiT) which can simultaneously learn the genre and instance features from the style image and then migrate the content and style representations into the stylized image. Compared with

[4, 28, 32] which capture the artist-style and artwork-style representations, our method ensures that style features are consistently and coherently distributed across the image, effectively maintaining the content image's main outline and texture specifics.

The overall framework is illustrated in Fig. 2. Our DGiT consists of three key components: content encoder, dual-head style encoder, and style transformer decoder. We first extract the content features $F_c$ and style feature $F_s$ via the transformer encoder, followed by the genre-wise head and instance-wise head to extract the genre features $F_{gen}$ and instance features $F_{ins}$. Then, we employ a designed style transformer decoder to stylize the content feature according to two style representations. Finally, we obtain the final stylized image $I_g$ by using an upsampling decoder. To encourage the model to extract the style features effectively, we design two contrastive losses, genre contrastive loss and instance contrastive loss, to guide the dual-head encoder to obtain discriminative genre features and unique instance features. During the training process, the genre labels are provided to the genre-wise head to supervise the learning process. In the test, we can generate the stylized image using the trained model without the genre label.

## 3.2 Dual-head Style Encoder

The dual-head style encoder consists of a genre-wise head and an instance-wise head. The genre-wise head is designed to capture the common features of the art genre such as the overall feeling, while the instance-wise head is utilized to capture the unique features like colors, texture, and brushstrokes of the artwork.

Specifically, we split a style image $I_s \in \mathbb{R}^{H \times W \times 3}$ into patches and map these input patches by using trainable linear projection to obtain a sequence of patch embeddings in the shape of $L \times C$. Here, $L = \frac{H \times W}{m \times m}$ represents the sequence length, $C$ and $m$ denote the embedding dimension and the patch size, respectively. Similarly to Str2 [6], these feature sequences are fed into the transformer encoder [7] to capture more refined feature sequences $F_s$. Finally, $F_s$ is fed into the genre-wise head and instance-wise head to further capture the genre and instance features. Two heads have the same structure but independent parameters. Each head consists of a multi-head self-attention module and a feed-forward network. The multi-head self-attention module can be formulated as:

$$\text{MHA}(Q, K, V) = [\text{head}_1, \ldots, \text{head}_h] W^O,$$
$$\text{head}_i = \text{Att}(Q W_i^Q, K W_i^K, V W_i^V),$$
$$\text{Att}(Q, K, V) = \text{softmax}\left(\frac{Q K^T}{\sqrt{d_k}} V\right),$$

$W_i^Q$, $W_i^K$, and $W_i^V$ are the learnable projection matrices. $W^O$ is the output projection matrix. $d_k$ is the dimension of the key. The genre-wise head is described as follows:

$$F'_{gen} = \text{MHA}(F_s, F_s, F_s) + F_s$$
$$F_{gen} = \text{FFN}(F'_{gen}) + F'_{gen},$$

We apply the layer normalized after the attention block. Similarly, the style sequence $F_s$ follows the same calculation after being fed into the instance-wise head. In Section 3.4, we optimize the model by minimizing genre contrastive loss and instance contrastive loss to guide the dual-head encoder to learn genres and instance features.

## 3.3 Style Transformer Decoder

Some existing methods [6, 34, 35] utilize ViT model, which exhibits slow convergence in training procedure due to a large number of parameters [22]. In contrast, we adopt AdaIN [9] combined with multi-head self-attention to alleviate computational cost and accelerate convergence. The more detailed of the style transformer decoder is shown in Fig. 3.

Specifically, we input the content sequence $F_c$ and the instance-specific sequence $F_{ins}$ into AdaIN [9] layer. This process can be formulated as follows:

$$A' = \text{AdaIN}(F_c, F_{ins}) = \sigma(F_{ins}) \left( \frac{F_c - \mu(F_{ins})}{\sigma(F_c)} \right) + \mu(F_{ins}), \quad (1)$$

where $\mu(\cdot)$ and $\sigma(\cdot)$ denote the mean and standard deviation of the input tensor, respectively. The obtained $A'$ is then fed into a multi-head self-attention module to obtain the intermediate representation as follows:

$$f'_{ins} = \text{MHA}(A', A', A') + F_c,$$
$$f_{ins} = \text{FFN}(AdaIN(f'_{ins}, F_{ins})) + f'_{ins}. \quad (2)$$

Subsequently, we merge the genre features $F_{gen}$ with the obtained intermediate representation across $N$ layers to generate the final output $F_{cs}$ via the similar processes in (2). Finally, the output feature sequence $F_{cs}$ is fed into three layers CNN decoder to obtain the stylized image $I_g$. Each layer includes two $3 \times 3$ convolutional layers, a ReLU layer, and an upsample layer.

*Analysis:* As we have two style heads, an immediate challenge is encountered for the decoder design, that is, how to reasonably integrate content and style features to obtain the final stylized image. A reasonable approach is to first merge instance features with content features to preserve texture details in the stylized image. Then, applying genre characteristics ensures uniform and coherent application of style elements across the entire image. We also analyze different fusion strategies in detail in Section 4.3.

## 3.4 Style Contrastive Learning

3.4.1 *Genre Contrastive Loss.* To encourage the genre-wise head to capture the common features of the art genre, we propose a genre contrastive loss as an implicit measurement to capture the genre-wise features effectively. It is well known that artworks in the same art genre share notable similarities in their use of color, line styles, and compositional techniques, while those from different genres exhibit significant differences. Hence, we classify style images from identical genres as positive examples and those from varying genres as negative examples, and design a genre contrastive loss for capturing the holistic discriminative genre style features.

Specifically, for a set of $N$ randomly sampled sample/lable pairs $\{x_i, y_i\}$, where $i \in I = \{1, \ldots N\}$ be the index of the sample within the batch. Let $A(i) = I \backslash \{i\}$ be the index set of all samples except $i$. We define $f_{gen} = \{z_i\}_{i=1}^N \in \mathbb{R}^{N \times d \times C}$ as the output of the genre-wise head. We randomly choose $z_i$ as the anchor, $P(i) = \{p \in A(j) : y_i = y_p\}$ as the index set of all positive samples, and $|P(i)|$ is its cardinality. We take $A(i) \backslash P(j)$ as the index set of all negative samples. The $\mathcal{L}_{gen}$ is designed to maximize the similarity between the anchor and positive samples while minimizing the similarity between the anchor and negative samples. The genre contrastive

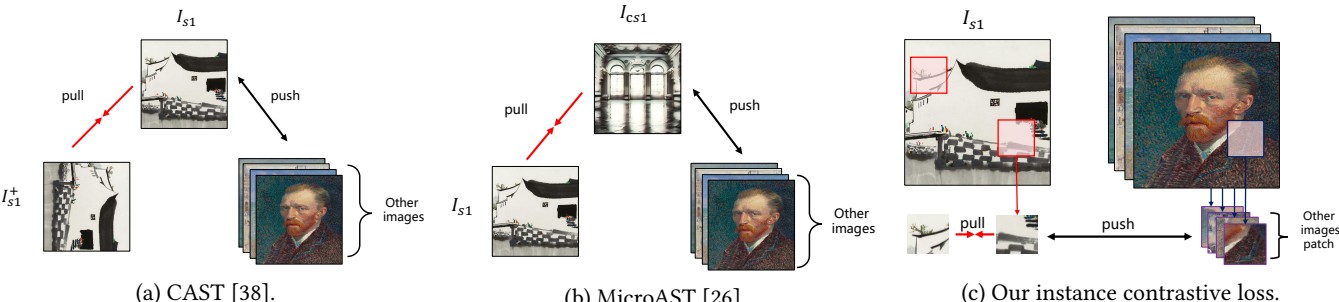

(a) CAST [38].

(b) MicroAST [26].

(c) Our instance contrastive loss.

**Figure 4: Illustration of contrastive loss designs in AST methods: (a) CAST [38] distinguishes between images by different artworks in general; (b) MicroAST [26] focuses solely on the relationship between the stylized image and its corresponding style image; (c) Our proposed instance contrastive loss exploit the texture details and brushstrokes across the arbitrary spatial locations of the style image.**

loss is formulated as:

$$\mathcal{L}_{gen} = -\frac{1}{N} \sum_{i \in I} \frac{1}{|P(i)|} \sum_{p \in P(i)} \log \frac{\exp(\hat{z}_i \cdot \hat{z}_p / \tau)}{\sum_{a \in A(i)} \exp(\hat{z}_i \cdot \hat{z}_a / \tau)}, \quad (3)$$

where $\hat{z} = f_1(z)$ is the nomalized output from the genre-wise head, $\cdot$ symbol denotes the innder product and $\tau \in \mathcal{R}^+$ is a temperature parameter. The function $f_1(\cdot)$ is a multi-layer projection that consists of two fully connected layers, to project the features into a $l_2$-normalized space.

*3.4.2 Instance Contrastive Loss.* To assist the instance-wise head in extracting the instance features, we design an instance contrastive loss to focus on the texture details and brushstrokes of the artwork. Some existing methods [26, 38, 39] leverage contrastive learning to enhance the network's proficiency in extracting style representations. As shown in Fig. 4, CAST [38] employs the style image $I_s$ and its augmented version $I_s^+$ as positive samples, with other N style images serving as negative samples. MicroAST [26] selects the stylized image $I_{cs_1}$ as the anchor, assigning its associated style image $I_{s_1}$ as a positive sample, and regarding other style images as negative samples. These methods process the entire image as an anchor to identify overall differences between various artworks. In contrastive, our instance contrastive loss focuses on the local textures and unique features at any spatial location within the style image. Different from patch-methods [17, 23] which considers the relationship between the stylized image and the source image, our loss can better assist the instance-wise head to extract the instance features by considering the relationship between a single artwork and other artworks.

We randomly select patches from the same style image as positive samples and patches from other style images as negative samples. This contrastive strategy enables the instance-wise head to focus on the distinct style variations in each artwork. In particular, we defined a set of $F_{ins} = \{v_i\}_{i=1}^N \in \mathbb{R}^{N \times d \times C}$ as the output feature of the instance-specific head. We randomly sample a pair of patches, $\hat{m}_i \in \mathbb{R}^{n \times c}$ and $\hat{m}_i^+ \in \mathbb{R}^{n \times c}$ on $v_i$ as positive sample while negative samples $\{m_j^-\}_{j \in A(j)}$ are sampled from the remaining $N-1$ images, maintaining the same shape as $\hat{m}_i$. Here, $n$ denotes the number of

the patch pixels.

$$\mathcal{L}_{ins} = -\sum_{i=1}^N \log \left( \frac{\exp(\hat{m}_i \cdot \hat{m}_i^+)/\tau}{\exp(\hat{m}_i \cdot \hat{m}_i^+)/\tau + \sum_{j \in A(j)} \exp(\hat{m}_i \cdot \hat{m}_j^-)/\tau} \right). \quad (4)$$

where $\hat{m} = f_2(m)$ is the nomalized output from the genre-wise head and $f_2(\cdot)$ is a multi-layer projection layer following the same structure of $f_1(\cdot)$.

## 3.5 Network Training

To preserve the original content structures and reference style patterns, we employ content perceptual loss to quantify differences between the generated image $I_g$ and the content image $I_c$, and style perceptual loss to measure the difference from the style reference $I_s$ to the generated image $I_g$.

We use feature maps extracted by a pre-trained VGG-19 [20] to calculate two losses. The content loss can be described as follows

$$\mathcal{L}_c = \frac{1}{N_l} \sum_{i=0}^{N_l} \|\phi_i(I_g) - \phi_i(I_c)\|_2, \quad (5)$$

where $\phi_i(\cdot)$ indicates the features map which is captured from the i-*th* layer of the VGG-19. $N_l$ denotes the number of layers. The style perceptual loss $\mathcal{L}_s$ can be described as follows

$$\mathcal{L}_s = \frac{1}{N_l} \sum_{l=0}^{N_l} \|\mu(\phi_i(I_g)) - \mu(\phi_i(I_s))\|_2 \\ + \|\sigma(\phi_i(I_g)) - \sigma(\phi_i(I_s))\|_2, \quad (6)$$

We also adopt our two contrastive objectives to learn genre-wise and instance-specific features from the style image. Hence, the total training objective function is formulated as

$$\mathcal{L} = \lambda_c \mathcal{L}_c + \lambda_s \mathcal{L}_s + \lambda_{gen} \mathcal{L}_{gen} + \lambda_{ins} \mathcal{L}_{ins}, \quad (7)$$

where the hyperparameters $\lambda_c$, $\lambda_s$, $\lambda_{gen}$, and $\lambda_{ins}$ are utilized to fine-tune the loss balance during the training.

## 4 EXPERIMENTS

### 4.1 Experiment settitngs

We utilize MS-COCO [16] and WikiArt [10] datasets as content and style images for the training process, where WikiArt dataset

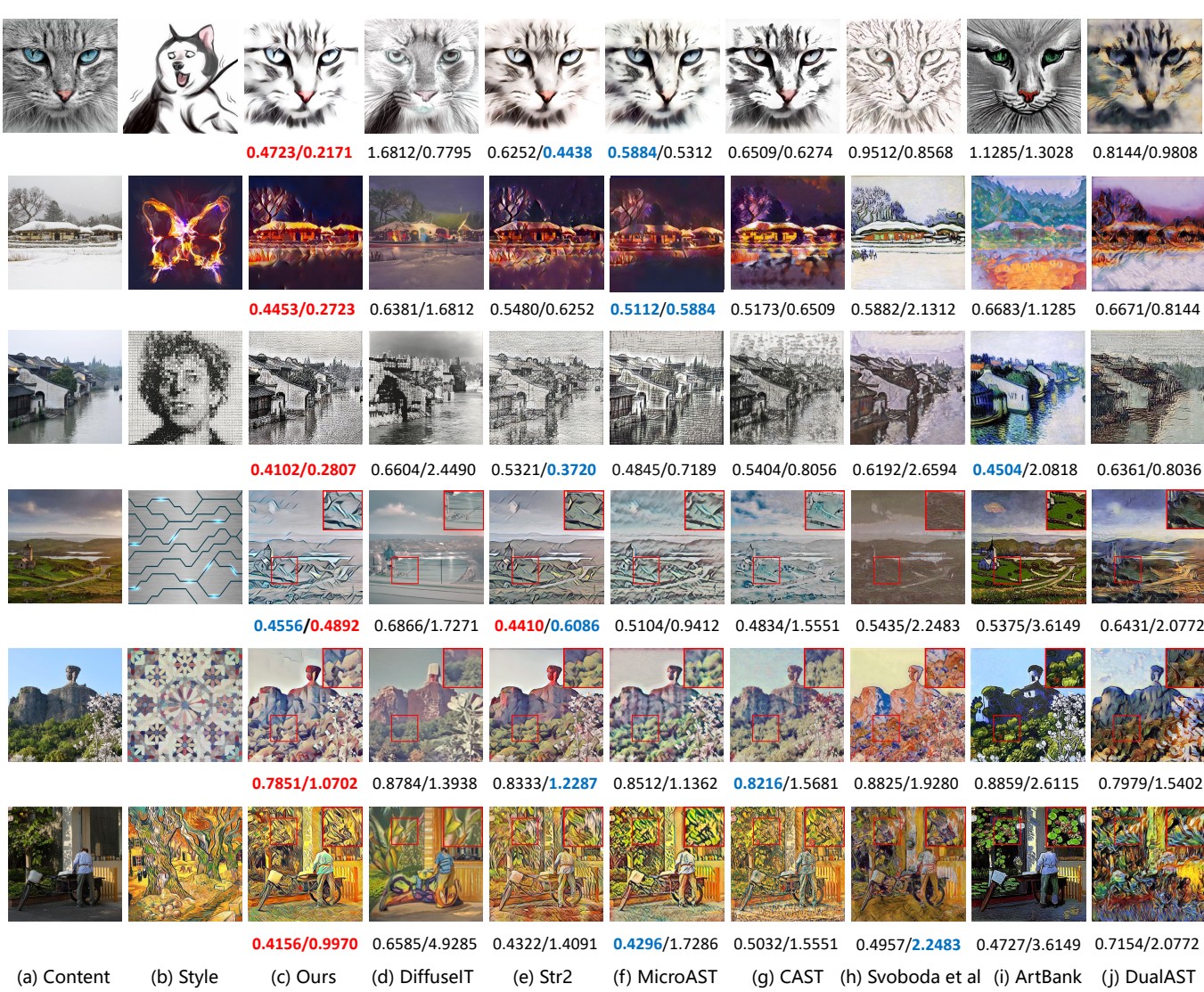

Figure 5: Qualitative comparisons. From the left to right: DiffuseIT [5], Str2 [6], MicroAST [26], CAST [38], Svoboda *et al.* [21], ArtBank [41], DualAST [4]. The scores are content/style losses. Red indicates the best score, blue is the second best.

Table 1: Quantitative comparison. Red indicates the best score, blue is the second one.

| Methods | Ours | DiffuseIT [5] | Str2 [6] | MicroAST [26] | CAST [38] | Svoboda *et al.* [21] | ArtBank [41] | DualAST [4] |
|---|---|---|---|---|---|---|---|---|
| SSIM ↑ | 0.5849 | 0.2487 | 0.5547 | 0.5420 | 0.2143 | 0.2652 | 0.3261 | 0.4783 |
| LPIPS ↓ | 0.4946 | 1.9687 | 0.5321 | 0.5232 | 0.6847 | 0.5839 | 1.2076 | 0.6672 |
| Style Loss ↓ | 0.5714 | 3.0719 | 0.6845 | 1.2302 | 1.1829 | 2.0416 | 3.8581 | 1.3476 |
| Content Loss ↓ | 0.9731 | 3.0312 | 1.1218 | 1.0986 | 1.0609 | 2.3811 | 2.6591 | 1.6407 |
| Style Pref.(%) ↑ | 27.44 | 2.52 | 23.64 | 14.08 | 13.8 | 5.2 | 1.56 | 11.76 |
| Content Pref.(%) ↑ | 26.48 | 2.72 | 15.72 | 17.56 | 17.12 | 4.4 | 3.88 | 12.12 |
| Overall Pref.(%) ↑ | 28.56 | 2.04 | 23.12 | 13.4 | 14.36 | 3.52 | 1.04 | 13.96 |

encompasses 28 artistic genres such as Impressionism, Ukiyo-e, Abstraction, etc. We utilize the Pikip dataset [38] as our testing dataset. Note that there is no overlap between this dataset and our training set, and the test images remain unlabeled. The training image is first resized to the dimension of $512 \times 512$ before being randomly cropped to $256 \times 256$ as inputs. The overall framework is

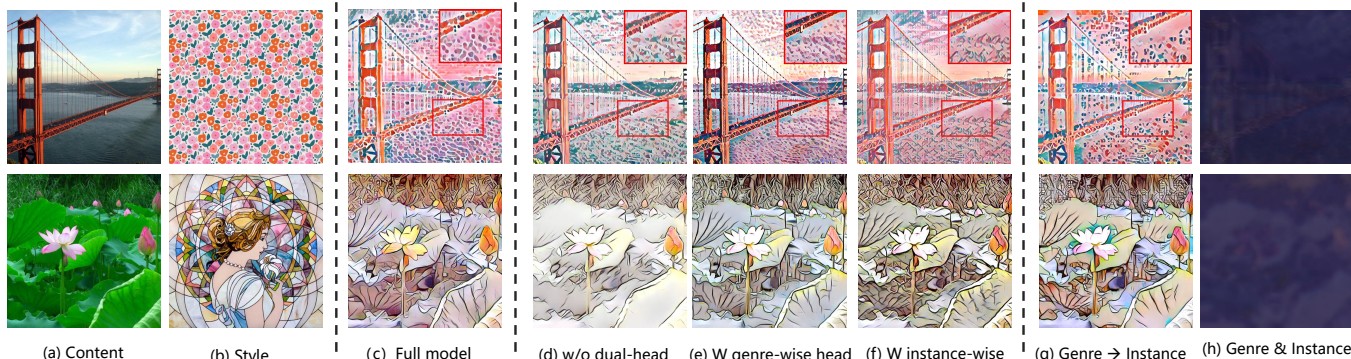

**Figure 6: Ablation results. (d)-(f) illustrate the different model configurations, respectively. (g) and (h) shows the results for different fusion strategies.**

implemented by Pytorch. We utilize Adam [12] as the optimization solver and the learning rate is $5e^{-4}$ using the warm-up adjustment strategy [30]. The network is trained for 160K with a batch size of 6. The loss weights are chosen as $\lambda_s = 10$, $\lambda_c = 7$, $\lambda_{gen} = 5$, $\lambda_{ins} = 1$. Our model can support any image resolution in the testing phase. All training experiments are developed on two NVIDIA RTX A5000 GPUs, and the testing data are developed on a single NVIDIA RTX A5000 GPU. More details are shown in the supplementary.

## 4.2 Comparing with State-of-the-Art Methods

We conduct comparison experiments with state-of-the-art algorithms on style transfer, including DiffuseIT [5], Str2 [6], MicroAST [26], CAST [38], Svoboda et al.[21], ArtBank [41], DualAST [4]. Among them, DiffuseIT [5], Str2 [6], MicroAST [26], CAST [38] learn the style representation from a single image, while Svoboda et al.[21], ArtBank [41] learn the style feature of the artist. DualAST [4] considers the both style of the artist and artworks. We evaluate the performance of these methods in terms of both qualitative and quantitative aspects. In the evaluation of all methods, we ensure a fair comparison by employing publicly available code and adhering to the default configurations provided for testing.

*4.2.1 Qualitative Evaluation.* To evaluate the superiority of our method, we compare our results with the seven methods above in Fig. 5. DiffuseIT [5], Str2 [6], MicroAST [26], CAST [38] learn the style representation from a single artwork. However, DiffuseIT [5] is unstable and tends to generate unrelated content structures (e.g. 2nd, 4th, and 5th rows). Str2 [6] and CAST [38] exist the problem of content leaks when the content image with complex structure (e.g. 4th, 5th, and 6th rows). MicroAST [26] often suffers from blurred and visual artifacts (e.g. 2nd, 3rd, and 4th rows). Svoboda et al. [21] and ArtBank [41] only capture the overall style of artists, limiting their ability to handle unseen artists and maintain style consistency with reference style images. As shown in (h) and (i), most results generated by the two methods lose the diversity of the style pattern. DualAST [4] considers the style of both artists and artworks, but the generated images often fail to capture the detailed texture and brushstrokes of the style image (e.g. 1st, 3rd, and 4th rows).

**Table 2: Quantitative comparison on ablation study. Red indicates the best score.**

| Methods | SSIM ↑ | LPIPS ↓ | Style Loss ↓ | Content Loss ↓ |
|---|---|---|---|---|
| w/o dual-head | 0.4321 | 0.6174 | 1.1761 | 1.3502 |
| w genre-wise head | 0.4973 | 0.5480 | 0.8154 | 1.1332 |
| w instance-wise head | 0.5301 | 0.5274 | 0.6568 | 1.2412 |
| genre+instance | 0.5638 | 0.5321 | 0.7363 | 1.1063 |
| instance&genre in parallel | 0.1201 | 0.7478 | 5.8871 | 1.7432 |
| full model | 0.5849 | 0.4946 | 0.5714 | 0.9731 |

Benefiting from dual-head style encoder, compared with DiffuseIT [5], Str2 [6], MicroAST [26], and CAST [38], our approach can capture both vivid local stroke characteristics and the overall appearance, while maintaining the content's structural integrity. Even when the content image contains complex structures, our method effectively transfers the rich texture of the style image to the stylized image while preserving its structural integrity. For example, as shown in the 1st-3rd row of Fig. 5, we can achieve the highest score in terms of content and style loss. It indicates our ability to capture a holistic similar style pattern and achieve reasonable texture mapping with the reference style image. In rows 4 to 6, our method transfers the color and texture of the style image to the stylized image while maintaining the structural consistency of the content image. In contrast to Svoboda et al. [21], ArtBank [41], and DualAST [4], we generate more diverse and vivid stylized images across content-style pairs from different artists and artworks.

*4.2.2 Quantitative Evaluation.* Table 1 shows the quantitative comparison with the above methods. We generated 10,125 stylized images by randomly selecting 75 content images and 135 style images. We calculate the SSIM [24] to assess the stylization quality with respect to content preservation. LPIPS loss [36] is adopted to measure the content fidelity between the content image and stylized images. Additionally, we evaluate the generated image's consistency in style and content with respect to the style and content references using style and content loss, respectively. As we can observe in Table 1, our method obtains better scores in all metrics. It indicates that our method exhibits superior capability in preserving both finer details and content affinity.

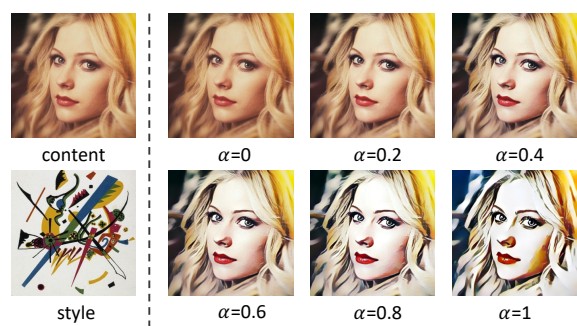

**Figure 7: Content-style trade-off by different $\alpha$ values.**

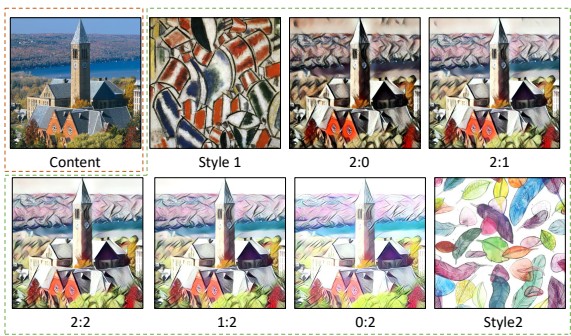

**Figure 8: Visualization of style interpolation.**

*4.2.3 User Study.* We compare our method with recent state-of-the-art methods to measure which method can generate reasonable images that are most accepted by humans. The users in our study include 25 males and 25 females, spanning the age range from 16 to 60. For each user, we randomly select 50 content-style pairs and present stylized results by our method and other methods in a randomized order. We ask users to evaluate results from the following three aspects: (i) which result exhibits superior style patterns. (ii) which result preserves the content structures more effectively, and (iii) which stylization result appears more natural and reasonable. Finally, we collected 2,500 votes from 50 users. We report the percentage of votes for each method in Table. 1. Our method receives significantly higher preferences in terms of content preservation, natural appearance and style consistency.

### 4.3 Ablation Study

*4.3.1 Analysis effective of dual-head style encoder.* We conduct an ablation study to evaluate the effectiveness of the dual-head encoder. From Fig. 6 (d) to (f), it illustrates the different model configurations without the dual-head style encoder, with only the genre-wise head and only the instance-specific head, respectively. We observe that the model in (d) generates images with less vivid texture and brushstrokes, and the style patterns are not fully transferred to the stylized image. We observe that the model in (d) generates images with less vivid texture and brushstrokes, and the style patterns are not fully transferred to the stylized image. In (e), the model might overlook the holistic style characteristics of artworks, leading to

content leakage and inconsistent local style textures. In (f), the model concentrates solely on local textural details and disregards the global style pattern, resulting in less appealing stylization with noticeable artifacts. The full model in (c) with the dual-head style encoder preserves the style consistency of the overall image and maintains the local detailed texture and specific brushstrokes of the artworks. The quantitative results are shown in Table 2.

*4.3.2 Different fusion strategies.* We conducted ablation studies on different fusion strategies in the style transformer decoder. The genre-wise head is designed to capture common features of the art genre, including overall feelings such as global compositional elements and general textural patterns common to the same art genre. The instance-wise head is utilized to capture the unique features of the artwork, such as colors, textures, and brushstrokes of the specific artwork. To enhance the generality of the model, we first fuse instance features with content features to preserve texture details for the stylized image, and then apply genre characteristics to ensure that style elements are uniformly and coherently applied across the entire image. As shown in Fig. 6 (c), the stylized image of our model exhibits the nature and vivid texture of the style image while keeping the content structure. In contrast, if the opposite fusion strategy is adopted, as shown in Fig. 6 (g), the prominence of individual style features might be somewhat subdued by the earlier emphasis on genre features. In addition, we replace the merge with the fusion strategy such as the parallel and sum operation. The results are shown in Fig. 6 (h), and the merge strategy is extensively corrupted, indicating a failure in model optimization.

### 4.4 Robustness Analysis

*4.4.1 Content-Style Trade-off.* The content-style trade-off aims to adjust the stylization intensity by changing the weight parameter $\alpha$. When $\alpha$ is increased to 1, we achieve complete stylization. As illustrated in Fig. 7, we produce a group of images showcasing gradual changes in style intensity. Our method can generate a wide range of stylization effects by adjusting $\alpha$ parameter, allowing users to customize the stylization intensity according to their preferences.

*4.4.2 Style Interpolation.* The style interpolation allows for interpolating multiple distinct style patterns into one generated image. We show the generated results under our method with different interpolations as exhibited in Fig. 8. Our method flexibly handles the interpolation between multiple styles. Users can combine various styles based on their preferences, resulting in personalized and diverse outcomes that cater to their individual needs.

### 5 CONCLUSION AND FUTURE WORK

In this paper, we propose a dual-head genre-instance transformer (DGiT) framework to simultaneously learn the genre and instance features from the artworks. Moreover, we propose two contrastive losses to enhance the ability of each head to extract the corresponding style. Our method not only generates detailed texture and specific brushstrokes but also ensures that style elements are uniformly and coherently applied across the stylized image. Extensive experiments demonstrate that our method can achieve the best style performance regarding visual quality. In the future, we will explore the application to the video-style transfer task.

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
