# OpenReview forum: "Dual-head Genre-instance Transformer Network for Arbitrary Style Transfer"
_acmmm.org/ACMMM/2024/Conference — MM2024 Poster_

### Official Review · Reviewer_C9dC · 2024-05-15

**Rating:** 3
**Confidence:** 3

**Summary:**

The fundamental insight of this paper lies in the recognition that artistic genres possess greater generalizability and adaptability compared to overarching artist-specific features. Consequently, this paper introduces the Dual-head Genre-instance Transformer (DGiT) framework, a novel approach that concurrently captures genre-specific and instance-specific features for arbitrary style transfer. This paper marks the first attempt to integrate genre and instance features in order to generate stylized images of unparalleled quality.

Furthermore, this paper devises two contrastive losses, which significantly enhance the network's capability to capture both style features. the methodology of this paper ensures a uniform distribution of the overarching style across the stylized image, while simultaneously enriching the details of textures and strokes in localized regions. Extensive qualitative and quantitative experiments demonstrate that the proposed method exhibits superior performance in terms of both visual fidelity and computational efficiency.

**Strengths:**

(1) The paper well-written. It is easy for readers to understand the core idea of this paper.

(2) The performance is good. The proposed method can achieve satisfactory qualitative and quantitative results.

(3) The proposed contrastive losses are interesting, which may inspire many other works.

**Limitations:**

(1) I think the concept of "genre" in this paper is similar to the concept of "domain", while the concept of "artwork" i similar to "exemplar". In the field Image-to-Image translation, there exist many works that translates images from one domain to another domain, for example, translating an image from photo domain to Ukiyo-e domain. An exemplar image can be used to control the translated results.

(2) All the visual results shown in this paper is taking the photo as the content image. It is better to show the visual results of transferring the style from photo to painting, that is to say, taking the photo as the style image.

(3) The designed Dual-head Encoder is somewhat incremental, which is simply a combination of vision transformer and AdaIN.

(4) Lack of experiments on contrastive learning.

**Suitability:**

2

---

### Official Review · Reviewer_DhUL · 2024-05-27

**Rating:** 5
**Confidence:** 3

**Summary:**

This paper proposes a novel Dual-head Genre-instance Transformer (DGiT) framework for arbitrary style transfer. DGiT simultaneously captures the art genre and instance features. Furthermore, two contrastive losses are proposed to enhance the network's capability to capture style features. The superiority of the proposed method is validated through extensive experiments.

**Strengths:**

1. The motivation is clear and reasonable. The art genre possesses greater generality than the comprehensive features of a specific artist.
2. The architecture design of the proposed DGiT and two losses make sense to me.
3. Both visual results and metrics demonstrate the excellence of the proposed DGiT.

**Limitations:**

1. It would be better if the author could include more discussion about the following papers which are related to the proposed method.

(1) Hong K, Jeon S, Yang H, et al. Domain-aware universal style transfer[C]//Proceedings of the IEEE/CVF International Conference on Computer Vision. 2021: 14609-14617.

(2) Kwon G, Ye J C. Clipstyler: Image style transfer with a single text condition[C]//Proceedings of the IEEE/CVF Conference on Computer Vision and Pattern Recognition. 2022: 18062-18071.

(3) Hong K, Jeon S, Lee J, et al. Aespa-net: Aesthetic pattern-aware style transfer networks[C]//Proceedings of the IEEE/CVF International Conference on Computer Vision. 2023: 22758-22767.

**Suitability:**

3

---

### Official Review · Reviewer_aNCK · 2024-05-27

**Rating:** 4
**Confidence:** 3

**Summary:**

This paper presents an arbitrary style transfer method by extracting silt features and content features separately.
Style features are then fed into the style transformer decoder to generate the final results.
The approach is generalizable to arbitrary styles and achieved better results.
Ablation studies and user studies were also conducted to further demonstrate the effectiveness of the methods

**Strengths:**

The result is better both objectively and subjectively based off the metrics.

**Limitations:**

1. DiffuseIT performed so bad as reported in the Table 1, can you explain the reasons?
2. Also, feeding the features into the generator is common in the diffusion era, can you compare with some of those methods?
3. Can you share some of the higher resolution results, such as 1K or 2K?

**Suitability:**

3

---

### Official Review · Reviewer_r4oA · 2024-05-30

**Rating:** 4
**Confidence:** 3

**Summary:**

This work proposes a dual-head genre-instance transformer (DGiT) framework to simultaneously capture genre and instance features for achieving arbitrary style transfer. The paper uses two contrastive losses to encourage DGiT to capture two style representations.

**Strengths:**

1. As stated in the paper, this work is the first to use art genre in the style transfer task. Numerous visual results demonstrate that it can achieve better performance than other structures.
2. Detailed experiments show the advantages of this method.
3. The results demonstrate the efficiency of the two contrastive losses, with quantitative comparisons supporting this.
4. By fusing two style features, this method can generate an image that combines both styles, which is interesting and has great potential.

**Limitations:**

1. More analysis of genre labels: The paper firstly introduces genre features to style transfer. However, more analysis of genre labels is needed. Specifically, while the paper mentions that there is no overlap between the test dataset and the training set, it is unclear if this situation is similar with genre information.
2. Generalization of the style loss: The method is intended for Arbitrary Style Transfer (AST), but its generalization to genre and content in the real world.
3. Some confuse of figure: In first row of Figure 6, in intuitions, I think results (e) maybe better preserved the instance information. Especially in structure of the bridge. Maybe more analysis in this situation will be better.
4. Unfair comparison: this paper uses two contrastive losses to train the network. While in quantitative comparison, these losses are used for comparison. This comparison is unfair for other methods that maybe not train with this target. In quantitative comparison, use other SOTA image feature extractor like DINOv2, which also different with the feature extractor of this method can be better.

**Suitability:**

3

---

### Meta-Review · Area_Chair_V5ww · 2024-07-01

**Recommendation:** Accept (Poster)
**Confidence:** 5

**Metareview:**

This paper introduced genre features for better image style transfer. Convincing qualitative and quantitative evaluations are presented.
All the reviewers are positive about this paper after the authors' rebuttals.